# Seismic Signaling for Detection of Empty Tunnels in the Plateau Zokor, *Eospalax baileyi*

**DOI:** 10.3390/ani13020240

**Published:** 2023-01-09

**Authors:** Kechi Dong, Jianwei Zhou, Feiyu Zhang, Longming Dong, Bin Chu, Rui Hua, Limin Hua

**Affiliations:** 1College of Grassland Science, Gansu Agricultural University, Key Laboratory of Grassland Ecosystem of the Ministry of Education, Engineering and Technology Research Center for Alpine Rodent Pest Control, National Forestry and Grassland Administration, Lanzhou 730070, China; 2Institute of Grassland Research of CAAS, Key Laboratory of Biohazard Monitoring, Green Prevention and Control for Artificial Grassland, Ministry of Agriculture and Rural Affairs, Hohhot 010010, China; 3Southwest Survey and Planning Institute of National Forestry and Grassland Administration, Kunming 650031, China

**Keywords:** plateau zokor, occupation, seismic signal

## Abstract

**Simple Summary:**

Seismic communication plays a crucial role in the behavior of subterranean rodents, particularly solitary ones. Studying the seismic communication and occupation behavior of subterranean rodents (especially solitary animals) provides a theoretical basis for understanding the adaptive evolution of communication methods in burrowing rodents and explaining their population recovery. The plateau zokor (*Eospalax baileyi*), a solitary subterranean rodent species endemic to the Qinghai–Tibet Plateau, will usually occupy empty neighboring tunnels in order to extend its territory. Little is known, however, about the process of territorial occupation and the function of animal communication during this process. In this study, we show that plateau zokors generate seismic signals to detect empty neighboring tunnels and then occupy the tunnels to extend their territory.

**Abstract:**

There are considerable challenges involved in studying the behavior of subterranean rodents owing to the underground nature of their ecotope. Seismic communication plays a crucial role in the behavior of subterranean rodents, particularly solitary ones. The plateau zokor (*Eospalax baileyi*), a solitary subterranean rodent species endemic to the Qinghai–Tibet Plateau, will usually occupy empty neighboring tunnels in order to extend their territory. Little is known, however, about the process of territorial occupation or the function of animal communication when occupation is taking place. Based on previous studies of subterranean rodent communication, we hypothesized that plateau zokors use seismic signals to detect neighboring tunnels and then occupy them when it was found their neighbors were absent. To test this, we placed artificial tunnels close to active original zokor tunnels to simulate the availability of an empty neighboring tunnel, and then the seismic signals when a zokor chose to occupy the empty artificial tunnel were recorded. The results showed that the frequency of zokors occupying artificial empty tunnels within 48 h was 7/8, In all of these instances, the zokors generated seismic signals before and after occupation of the empty artificial tunnel. The number of seismic signals generated by the zokors increased significantly (*p* = 0.024) when they detected and occupied the artificial tunnels, compared to those generated in their original tunnels without the presence of an artificial tunnel alongside. Inside the original tunnels, the inter-pulse time interval of the seismic signals was significantly higher (*p* < 0.001), the peak frequency of these signals was significantly higher (*p* < 0.01), and the energy of the signals was significantly lower (*p* = 0.006), compared with those when an artificial tunnel was positioned next to the original. The results of this study suggest that plateau zokors first generate seismic signals to detect empty neighboring tunnels and that they are empty. In the absence of neighbor plateau zokors, they occupy the empty tunnels to extend their own territory.

## 1. Introduction

Territoriality is a survival strategy used by many animals [1] to compete for resources [2,3]. Conspecifics usually occupy territories adjacent to, or take over parts of, their neighbors’ territory for the purpose of exploiting resources and providing mating opportunities [4]. Different species use a wide variety of tactics to scout the potential and available territory, such as use of auditory, visual or olfactory signals [5]. Studying the mechanisms of animal incursion behavior is very important for a better understanding of their survival strategies.

Animal communication is defined as the transfer of information from one individual to another, intraspecifically or interspecifically, through different channels, for the purpose of increasing the overall fitness and chances of survival not only of the individual but also of the population [6]. Communication may help animals find mates, coordinate group behavior, and occupy or defend their territory [7]. For aboveground mammals, visual and olfactory cues are the usual means of communication between and within species [8,9]. However, there are big challenges in studying the communication of underground animals because of the unique nature of this ecotope. 

Subterranean rodents are small mammals that live in underground tunnels and perform most of their life activities, such as foraging and breeding, in their tunnel system [10]. Across the globe, there are at least 140 subterranean species belonging to 20 genera of 8 families, and they can be found in all continents except Australia and Antarctica [11]. These mammals are adapted exclusively for underground life, involving morphological and behavioral specializations [12]. Almost 50% of subterranean rodents are solitary species [13]. These species are those that live singly in their own tunnel system and meet only for courtship and mating during the breeding period [14]. The tunnels of solitary subterranean rodents are independent and separated by soil. Within a family comprising females and their pups, auditory, visual or olfactory signals can be used to communicate information [15]. However, when these animals live in their isolated tunnels, in particular within the non-breeding season, they must adapt to and evolve special ways to communicate to adapt to life under the ground [14]. Compared with above ground, the underground environment of subterranean rodents, including its total darkness, may exert selective pressure on certain phenotypic traits leading to degradation of the visual system [16]. Solitary subterranean rodents seldom use olfactory or auditory cues for communication between adult individuals because of the isolation of their tunnels. Transmission of olfactory and auditory signals through the soil between tunnels is extremely challenging because these signals attenuate rapidly with distance. This means that the distances over which these communication modalities can be effectively used is severely limited [17,18]. Therefore, the usual means of sensory communication for occupying or defending territory in subterranean rodents is not completely understood.

Studies indicate that subterranean rodents use their feet, head, snout and teeth to tap the walls of tunnels to produce relatively weak seismic signals [14]. More specifically, among 23 species of subterranean rodents, it was found that 74% used their feet, 13% their front teeth, and 13% their head, snout and chest [13]. Solitary subterranean rodents generate seismic signals from specific parts of their bodies and the information can be communicated across adjacent and sometimes relatively distant tunnels [19]. This communication method, which uses body parts to knock on the environment matrix to transmit small shock waves to realize information exchange between individuals, is defined as seismic communication [20]. Previous studies have identified two families and six species of subterranean rodents that use seismic communication for territory protection [21,22], including three species of Bathyergidae (the Namaqua dune mole-rat (*Bathyergus janetta*), the Cape dune mole-rat (*Bathyergus suillus*), and Cape mole-rat (*Georychus capensis*)) and three species of Spalacidae (the Gansu zokor (*Eospalax cansus*), the Demon African mole-rat (*Tachyoryctes daemon*), and the blind mole-rat (*Spalax ehrenbergi*)) [13]. 

The plateau zokor (*Eospalax baileyi*) is a solitary subterranean rodent endemic to the alpine grassland ecosystems of the Qinghai–Tibet Plateau (QTP) [23,24]. These animals are known as “ecological engineers” because they play a vital role in the maintenance of diversity of plant species and the food web, and in soil nutrient cycling [25,26]. However, zokors are considered as grassland pests because of the many mounds that they create, which can kill plants by covering them, and cause soil erosion, resulting in alpine rangeland degradation [27]. Fan et al. [28] found that the empty tunnels left behind by plateau zokors either killed by man-made traps or having died naturally were quickly occupied by zokors living in adjacent tunnels. The new occupant would take over the empty tunnels as part of their existing tunnel system and enjoy the food left in the empty tunnels. Clearly, such behavior provides considerable benefits for both the individual invader and the broader population in terms of quickly recovering its size. This suggests that some form of intraspecies communication may also play a role in dissuading and warding off intruders during the time when a zokor is alive and regularly using its tunnel. So the question is how do plateau zokors know that a tunnel has been abandoned and when to take over a neighboring tunnel without triggering aggression that may otherwise ensue when attempting occupation of a neighboring tunnel?

Based on previous behavioral studies of subterranean rodents [29], we hypothesized that plateau zokors routinely use seismic signals to detect neighboring tunnels. If they do not receive a signal in return, a zokor will dig out a way to enter the empty tunnel and occupy it. To test this hypothesis, we conducted a field trial in an alpine meadow in Tianzhu County in the northeastern QTP. Artificial chambers were built in locations close to active zokor tunnels to simulate a neighboring empty tunnel and the seismic signals generated when a zokor attempted an occupation were recorded. The aim was to provide novel insights into territory occupation strategy and associated behaviors in plateau zokors in the QTP region.

## 2. Materials and Methods

### 2.1. Study Site and Animals

The study site was located in Tianzhu Tibetan Autonomous County on the northeastern edge of the QTP (37°12′13″ N, 102°46′11″ E; altitude: 2957 m) [26] (Figure 1). The type of grassland in the study site was alpine meadow, with 90% vegetation coverage. The average annual temperature is −0.1° and the annual precipitation is 416 mm. The dominant plant species are *Kobresia humilis*, *Elymus dahuricus*, *Kobresia capillifolia*, *Potentilla anserina*, *Gentiana macrophylla*, and *Medicago ruthenica*. The plateau zokor is the only subterranean rodent in the study site.

### 2.2. Locating the Original Tunnels

A 4.65-ha alpine meadow with new zokor mounds was selected for this study in July and August 2021, which is the non-breeding period for plateau zokors and the non-grazing season for sheep and yaks. A steel probe (0.8 cm in diameter, 90 cm in length) was used to locate original active tunnels near to new mounds of plateau zokors [30], which were then marked with different colored flags for ease of identification during the trial. The probe was also used to find straight sections of original tunnels alongside which the artificial tunnels could be built. In total, the natural tunnels of eight zokors were selected and regarded as the “original tunnels”. The distance between them was more than 100 m, meaning they were well separated.

### 2.3. Construction of the Artificial Tunnels

An auger drill was employed to build a sealed artificial tunnel for simulating an empty zokor tunnel (Figure 2). The length of the artificial tunnel was 70 cm. It had the same diameter (10 cm) and depth (15 cm below the soil surface) as the original tunnels. In the natural environment, the effective acquisition distance of the plateau zokor seismic signal is about 2 m, whereas in the indoor environment, the plateau zokor seismic signal is collected every 20 cm, Therefore, this study chooses the field acquisition seismic signal distance as 20 cm to meet the acquisition requirements [31].

### 2.4. Recording of Seismic Signals in the Original Tunnel and during Occupation

A custom seismic signal acquisition system including a bio-signal acquisition and analysis system (BL-420F, China; sensitivity: 23.4 ± 7.5% V/M/S; sampling frequency: 1000 Hz), a laptop, and storage battery (12 v, 70 Ah) were used to record the seismic signals produced by zokors in the original tunnels without the presence of artificial tunnels as the control (CK), as well as the signals when zokors passed by the parallel artificial tunnels as the experiment (EX). Four seismometer (LGT-20 4.5 Hz, China) were positioned in line at a distance of 5 cm on both sides of the original tunnels for recording the seismic signals of plateau zokors (Figure 2a). Among the four seismometers (8 Level, Wavelet 1-D, one-Dimensional), three were placed on both sides of the original tunnel, 5 cm away from the original tunnel, and one was placed on the right side of the artificial tunnel, 5 cm away from the artificial tunnel (Figure 2b). The generation time and number of seismic signals were recorded, recording the full 48 h between the scientists leaving the area and returning to dig up the tunnels (Figure 2d).

We used the seismic signals to judge the zokor’s location, distance and locomotion in their original tunnels. Firstly, we placed two seismometers on each side along the original tunnels before we dig the artificial tunnel. The seismic signal detection range of the seismometer was 0~2 m. When a zokor move to the seismometer closely, the detected signals were stronger. So, the seismic signals were recorded by these seismometers for judging the location and the distance between the zokor and the seismometer. After we built the artificial tunnel paralleled the original tunnel, the zokor would spend more time in the place inside the original tunnel and produced more seismic signals in the place. Therefore, we judged that the zokor passed by the artificial tunnel next to the original tunnel.

To avoid disturbance by human movement on the pasture, the seismic signals recorded after the researchers had left the study site were selected.

### 2.5. Counting Occupations of the Artificial Tunnels

It was supposed that a zokor would find an empty tunnel using seismic signals and then occupy it, which we judged by opening and checking the artificial tunnels after 48 h. If the artificial tunnel was connected to the original tunnel, this was defined as a successful occupation. If the artificial tunnel was filled with soil pushed by the zokor, it was defined as accomplishing a defense of the tunnel. The numbers of occupied and defended artificial tunnels were counted.

### 2.6. Definition of Measurement Parameters Related to Seismic Communication in the Study

Several parameters were used in the analysis of the signals: (a) pulse-thump; (b) pulse group (bouts)-sequential pulses; and (c) series, the number of bouts. Six basic variables were used to characterize the signals: (1) the number of thumps (pulses) in a bout; (2) the duration of the bout; (3) the number of bouts in a series; (4) the duration of the pulse; (5) peak frequency; and (6) energy. We define all terms as follows (Table 1):

### 2.7. Data Analysis

The number of activities, the duration of seismic signals, inter-bout distance, bouts per series, pulses per bout, inter-pulse distance, peak frequency and corresponding peak frequency energy of seismic signals were compared between control and experimental situations.

SPSS 17.0 was used to statistically analyze the experimental data. The data were first tested for a normal distribution (Kolmogorov–Smirnov test) and homogeneity of variance (Levene’s test). Matlab 2014a was used to obtain the fast Fourier transform in terms of the range of the seismic signal time and frequency and seismic signal correlation parameters. The obtained seismic signal correlation parameters, activity counts, and seismic signal duration data were tested for differences between seismic signals using a non-parametric test (Mann–Whitney U test) with a significance threshold set at *p* = 0.05 (95% confidence interval). Origin Po (2022b) was used to plot the plateau zokors’ seismic signals. Matlab 2014a was used to plot their vibration, time and frequency domains.

## 3. Results

### 3.1. Proportion of Occupied Tunnels among All Artificial Tunnels

The results showed that seven artificial tunnels were occupied by plateau zokors after 48 h, accounting for 7/8 of the total number of artificial tunnels. Among these occupied tunnels, 5/7 were connected to the original tunnel as part of the zokor’s tunnel system, with the remaining 2/7 having been filled with soil by the zokor that found the artificial tunnel. All the original tunnels adjacent to the artificial tunnels remained intact.

### 3.2. Number of Activities and the Duration of Seismic Signals Produced by Zokors in the Original Tunnel and When Artificial Tunnel Exists

According to whether there are artificial tunnels at 20 cm next to the eight original tunnels, the 48 h seismic signals of whether there are artificial tunnels are collected, respectively, and combined with seismic signal analysis the number of activities and duration of seismic signals within 48 h for plateau zokors(n = 8). Significant differences were found in the number of activities and the duration of seismic signals produced between the zokors in the original tunnels and when they perceived by the artificial tunnels (Figure 3). Specifically, plateau zokors increased their activity numbers and produced more seismic signals when they perceived by the artificial tunnels.

### 3.3. Differences in the Parameters of Seismic Signals of Plateau Zokors Recorded in the Original Tunnels and When Perceived by Artificial Tunnels

The original tunnel continuously collects seismic signals for 48 h, and the artificial tunnel continuously collects seismic signals for 48 h. A total of 92 series of seismic signals produced by plateau zokors were recorded, including 6 series of seismic signals in the original tunnels and 86 series of seismic signals in the artificial tunnels. After inputting the recorded signals into Matlab, vibration signal, amplitude FFT—Spectrum plots of the signals were obtained under the two conditions (Figure 4 and Figure 5).

The seismic signals were compared between control and experimental situations, it was found that the number of percussion rounds, the time interval between adjacent rounds, and the number of pulses per round of seismic signals when after occupation of the empty artificial tunnel, were greater than those made in the control experiments without experimental tunnel (Figure 6). The time interval between pulses of the seismic signals produced in the control experiments without experimental tunnel was significantly greater than in the artificial tunnel (*p* < 0.001, Figure 6). In addition, the peak frequency of the seismic signal produced by plateau zokors in the original tunnel was significantly higher than that in the experiments with artificial tunnel (*p* < 0.001), and the corresponding peak frequency energy was significantly lower than that within the artificial tunnels (*p* = 0.06, Figure 7).

## 4. Discussion

### 4.1. Why Do Plateau Zokors Use Seismic Signals to Detect Neighboring Tunnels?

Previous studies have revealed that most solitary subterranean rodents use seismic communication to share information, such as *Spalax ehrenbergi, Bathyergus suillus* and *Dipodomys spectabilis* [13,14]. Low-frequency seismic signals can be transmitted much faster, and farther than airborne sounds between the separated tunnels of solitary subterranean rodents [29]. Nevo et al. [32] proposed that the seismic channel might represent a useful medium for communication among subterranean rodents in general. Seismic sensitivity might also be useful for protecting territory, foraging and courtship. 

The plateau zokor is a typical solitary subterranean rodent in the QTP region [24]. Their courtship period is from March to May in early spring, after which they generally return to their own tunnels. However, Ji [33] found that the home ranges of plateau zokors overlap in June and July, which is the non-courtship period. Fan et al. [28] also reported that plateau zokors demonstrate occupation behavior in the field in autumn. However, they did not explain the mechanism of how such occupation behavior occurs.

In previous work published by our group, it was found that the plateau zokor produced seismic signals both in the laboratory and in the field. The transmission distance of the seismic signals was around 2 m in the field; this is given as the ambient seismic noise, the distance at which plateau zokor signals are no longer clearly detected [31,34]. It refers to the distance beyond which it is difficult to clearly detect the plateau zokor seismic signal under the given environmental seismic noise. In the non-breeding period, the territory of the zokor is generally 560 ± 101 m^2^ for males (n = 8) and 640 ± 130 m^2^ for females (n = 8) [35], with the distance between territories ranging from 2 m to 22 m [33]. Narins et al. [36] reported that the distance of seismic signal transmission between burrows of the Cape mole-rat (*Georychus capensis*) ranged from 3 m to 4 m. The Gansu zokor (*Eospalax cansus*) uses seismic communication for scoping out or protecting its territory [37]. The plateau zokor and the Gansu zokor belong to the genus *Eospalax* and their morphologies and behaviors are similar. 

### 4.2. What Is the Mechanism Employed by Plateau Zokors When Occupying a Tunnel?

Zhou et al. [31] and Dong et al. [34] reported that plateau zokors used their snout to tap the side wall of an artificial acrylic tunnel to generate seismic signals. In the present study, it was found that they produced seismic signals while patrolling inside the original tunnel, perhaps using the signals to announce their territory or scope out the territory of others [34]. Hegab et al. [38] reported that plateau zokors showed a strong preference for novelty and increased exploration time for novel objects, and were effective at remembering changes in locations of new objects. In this study, when zokors found a nearby empty tunnel, they increased their staying time at the location in the original tunnel close to the empty tunnel. This can be seen from the continuous generation of seismic signals by plateau zokors in the presence of artificial tunnels (Figure 3), and produced more signals (Figure 3). The inter-bout time interval, bouts per series, and pulses per bout of the seismic signals collected by the plateau zokor artificial tunnel are higher than those of the original tunnel, but the time interval between pulses in the original tunnels was significantly higher than that the artificial tunnels. In these situations, it might be the case that the zokor regarded the tunnel as a potential threat and sought to protect its own territory. So the 8th tunnel was not occupied. Of these artificial tunnels, 5/7 were connected with the original tunnel as part of the territory, whilst the remaining 2/7 were blocked by the invading zokor. In these situations, it might be the case that the zokor regarded the tunnel as a potential threat and sought to protect its own territory.

### 4.3. How Does the Plateau Zokor Perceive the Seismic Signals?

The reception of seismic signals in subterranean rodents may occur in either or both of the following ways: through their auditory and somatosensory systems. In the former, the auditory nerve receives the auditory signals directly through airborne transmission or indirectly by bone conduction, such as through the jaw or skull [39]. The outer ear of most subterranean rodents is degenerated and airborne signals cannot be transmitted over long distances, and even less so through soil. Some scientists have concluded that auditory signals are only used in social subterranean rodents or within a family in the same tunnel [40]. Bone conduction might be an effective channel of signal transmission in subterranean rodents, with some using this method to achieve seismic communication [41]. Besides airborne transmission, some subterranean rodents use their somatosensory system to perceive seismic signals, such as the lamellate corpuscle mechanoreceptors in the paws of the blind mole-rat (*Spalax ehrenbergi*) possibly being used to detect low-frequency seismic signals. The use of generated vibration signals (seismic “echolocation”) may be used to detect empty tunnels [29]. Although this study did not carry out an experiment on the reception of seismic signals in plateau zokors, Xu et al. [42] reported that the *Prestin* gene was expressed in the tissues of the cochlea, tail, footpad and snout of plateau zokors, which might potentially be helpful in the perception of low-frequency seismic signals and spatial orientation. The plateau zokor might use its somatosensory system for orientation in the new empty tunnel.

The present study reveals novel findings on the relationship between seismic communication and territorial invasion in plateau zokors. The current results support previous assertions that subterranean rodents are effective utilizers of substrate-borne vibrations generated for intraspecies communication owing to the limited sensory cues available underground. The results of this study may partly explain the built-in adaptive mechanism by which populations of plateau zokors recover when some individuals are removed from their habitat. Plateau zokors use seismic signals to encroach upon and expand their territory; this is conducive to the use of existing resources (food, tunnel space, etc., which in turn facilitates mating). Thus, the main ecological factors that determine the characteristics and evolution of animal mating systems may include the opportunistic availability of resources in addition to the regular distribution of food and nesting sites in time and space [43,44]. The timely availability of such resources appears to depend heavily on subterranean signaling, and the related sensory capabilities of animals.

## 5. Conclusions

In summary, plateau zokors seem to produce low-frequency seismic wave signals as part of the process involved in the occupation of empty tunnels. Moreover, when the artificial tunnels were present, the time animals took to explore their surroundings increased, suggesting that plateau zokors are well adapted to living underground. Seismic communication is well suited to underground solitary rodents trying to communicate over relatively short and possibly over long distances, but the species-specific mechanisms of signal generation and reception remain to be studied in future work.

## Figures and Tables

**Figure 1 animals-13-00240-f001:**
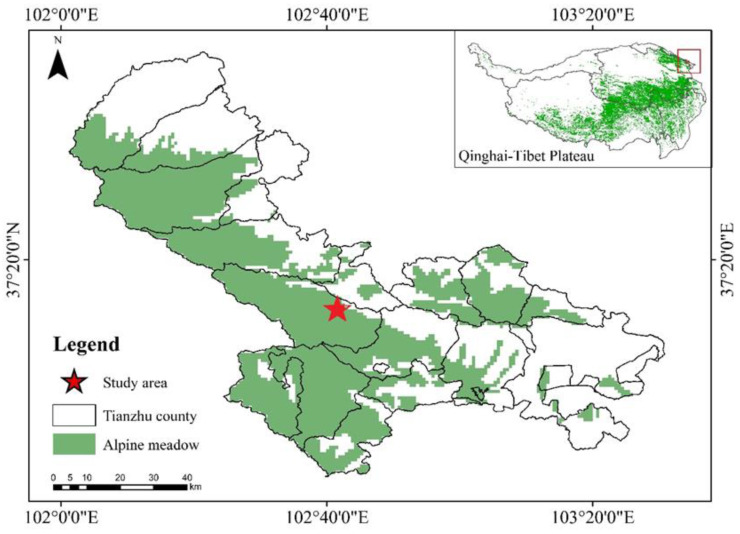
Location of the research site in Tianzhu Tibetan Autonomous County on the northeastern edge of the Qinghai–Tibet Plateau.

**Figure 2 animals-13-00240-f002:**
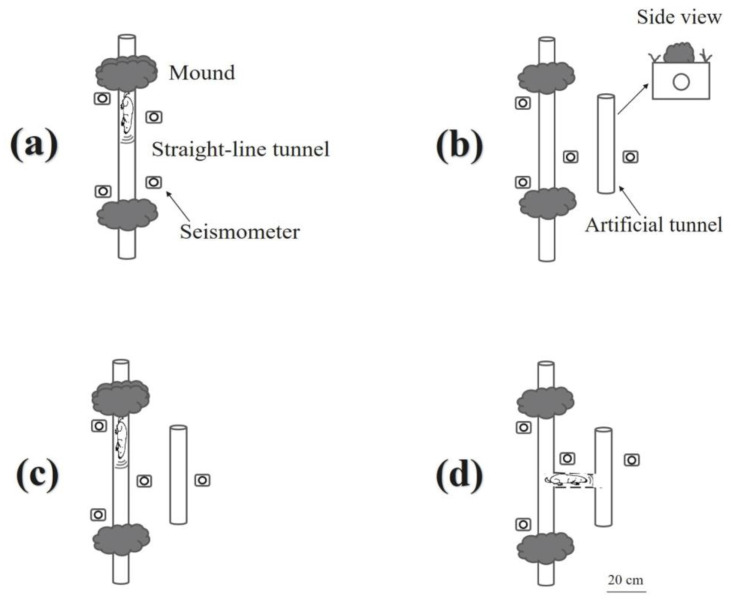
Schematic drawings of the stages of the occupation experiment with plateau zokors (plan view): (**a**) seismic signals near original tunnels; (**b**) artificial tunnels based on native plateau zokor activity; (**c**,**d**) acquisition of seismic signals during occupation of artificial tunnel.

**Figure 3 animals-13-00240-f003:**
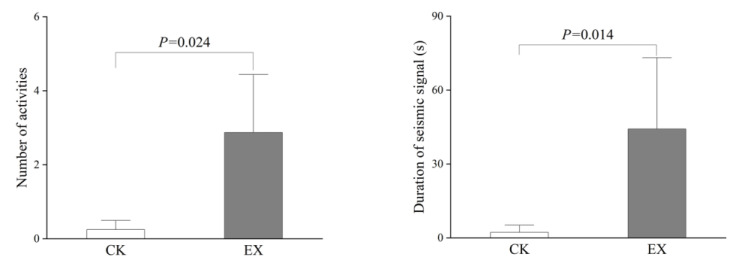
Comparison of the number of activities and duration of seismic signals within 48 h for plateau zokors in their control (CK) and experimental situations (EX). The bars in the figure are mean ± SE.

**Figure 4 animals-13-00240-f004:**
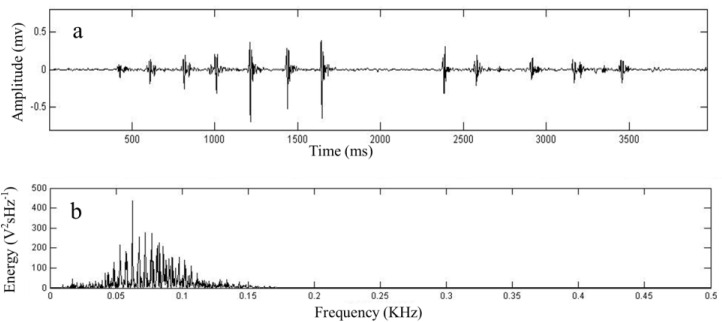
Seismic signal of plateau zokors in the control experiments without experimental burrows (**a**) and Seismic signal amplitude FFT-Spectrum plots (**b**).

**Figure 5 animals-13-00240-f005:**
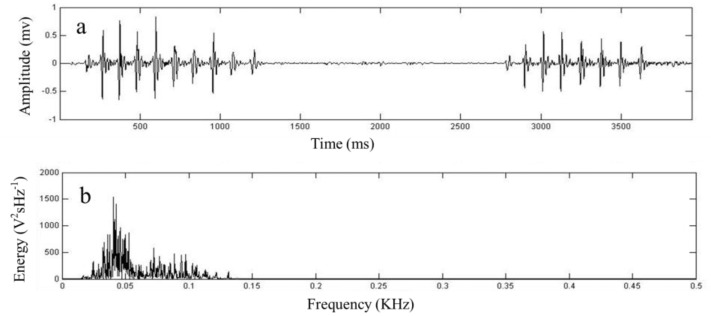
Seismic signal of plateau zokors in the experiments with artificial burrows (**a**) and Seismic signal amplitude FFT-Spectrum plots (**b**).

**Figure 6 animals-13-00240-f006:**
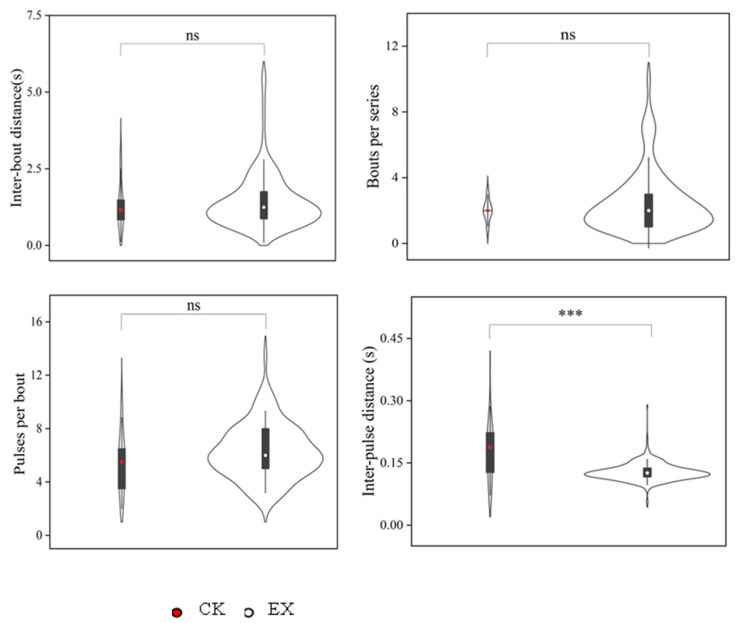
Comparison of the inter-bout time interval, bouts per series, pulses per bout, and inter-pulse time interval of the plateau zokor Seismic signal were compared between control and experimental situations. within the natural and artificial tunnels. In the violin plot, the lower and upper edges of the box represent the 25% (q1) and 75% (q3) quartiles, respectively. The dots inside the box represent the median (md). The whiskers extend to the most extreme values the dot inside the box, md ± 1.5 (q3 − q1). Note: *** *p* < 0.001; ns, non-significant. CK: natural tunnels; EX: artificial tunnels.

**Figure 7 animals-13-00240-f007:**
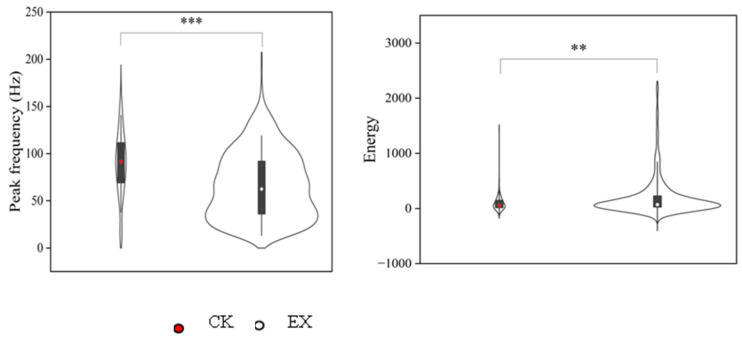
Comparison of the main peak frequencies and energies of the Seismic signals were compared between control and experimental situations. In the violin plot, the lower and upper edges of the box represent the 25% (q1) and 75% (q3) quartiles, respectively. The points inside the box represent the median (md). The whiskers extend to the most extreme values within inner fences, md ± 1.5 (q3 − q1). Note: ** *p* < 0.01; *** *p* < 0.001. CK: natural tunnels; EX: artificial tunnels.

**Table 1 animals-13-00240-t001:** Definition of measurement parameters related to seismic communication in the study.

Parameters	Explanation
**inter-pulse time interval**	The duration of the silence between two successive pulses.
**pulses per bout**	The number of thumps (pulses) in a bout.
**inter-bout time interval**	The duration of the silence between two successive bouts.
**bouts per series**	Number of bouts in a series.
**number of activities**	The number of the series.
**duration of seismic signals**	The duration of the series.
**peak frequency**	Seismic signal amplitude FFT—Spectrogram abscissa value.
**energy**	Seismic signal amplitude FFT—Spectrogram ordinate value.

## Data Availability

The datasets used and/or analyzed during the current study are available from the corresponding author on reasonable request.

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
