# Peer review of "Seismic Signaling for Detection of Empty Tunnels in the Plateau Zokor, Eospalax baileyi"

_animals, 2023, doi:10.3390/ani13020240_

Round 1

Reviewer 1 Report

This was a very interesting paper relating to how zokors appear to use seismic signals to identify neighbouring tunnels. The authors have conducted some difficult field experiments and have found some important new information about these enigmatic animals.

My major comments are the following:

1)     I found it very hard to tell exactly what the parameters were that were measured. The authors did not define their terms and seemed to use terms interchangeably. For example, on line 162 they list “generation time and number” and also “round number per series”. Later on, “number of activities” and “number of percussion rounds” are mentioned, then later still we read about “number of percussion sessions” and “pulses in a single session”. In Fig. 6, the terminology changed between graphs and caption. I don’t know if these are the same thing. For example, if we look at Fig. 4a, does this show 1, 2 or 12 “activities”? Is this 1 or 2 “rounds”? I really don’t know because no definitions are provided. The authors must define particular aspects of the seismic activity, using specific terms which are then used throughout the manuscript. They should also provide a diagram which labels all these parameters of the seismic signal.

2)     In several places, the authors seemed to be referring to the location of the zokors (e.g. recordings made “within the artificial tunnels”), and refer to distances, locomotion within tunnels (zokors “passing by”) and time spent exploring. However, it is not clear to me that the zokor’s location and activity, other than seismic signalling, was actually recorded in this study. If it was, how this was done should be clearly described. In some cases I think that these terms may be mistranslations. More clarity is needed throughout, and I have pointed out examples that need attention in my Minor Comments, below.

3)     The authors mix observations with interpretations multiple times in the Discussion, and also in the title of this manuscript. What they actually found, I believe, is that the pattern of seismic signalling changes when an artificial tunnel is dug next to a natural tunnel. Moving from there to saying that the seismic signals are used to “detect” these tunnels is a leap. I agree that the signals might potentially be used for detection of empty tunnels through some form of “seismic echolocation”. This has been described in Spalax (Kimchi’s work, listed below), but there is no mention of “seismic echolocation” in the current paper. Kimchi’s work should be described in the Discussion so the reader can see how seismic signals could in principle be used by the zokor. However, there is no direct evidence provided here to show that the zokor is doing the same: perhaps it has detected the new tunnel by another sensory modality, and the seismic signalling is aimed at a conspecific which the zokor thinks might live in that new burrow. We don’t know. The zokor certainly cannot be said to use the signalling to “occupy” the burrow. The title needs to be rewritten to reflect what was actually found here, and observations and interpretations need to be clearly separated in the Discussion.

4)     The Introduction does not say very much about the burrows and seismic signalling of zokors, but this would be useful to the reader in order to understand more about the animal being examined. A lot of what is needed is actually included in the Discussion, sections 4.1 and 4.2, sections which include very little new information from the present study. I therefore recommend that some of the material in that section (especially section 4.1, and also a mention of how the zokor generates seismic signals from 4.2) should be moved and intregrated into the Introduction, so the reader knows more about zokor behaviour before reading about this particular study. Then, the Discussion can be used to put the new results in context.

5)     All p-values should be reported throughout text and figures: see Amrhein et al. (2019) for widespread concerns about binary significance thresholds.

I have a number of minor comments, below, most of which should be easily addressed. I understand that it cannot be easy writing a paper in a different language to your own; I hope the comments I provide below are clear and helpful.

Amrhein, V., Greenland, S., McShane, B. et al. (2019) Retire statistical significance. Nature 567: 305-307.

Kimchi T, Terkel J (2003a) Detours by the blind mole-rat follow assessment of location and physical properties of underground obstacles. Anim Behav 66:885–891

Kimchi T, Terkel J (2003b) Mole rats (Spalax ehrenbergi) select bypass burrowing strategies in accordance with obstacle size. Naturwissenschaften 90:36–39

Kimchi T, Reshef M, Terkel J (2005) Evidence for the use of reflected self-generated seismic waves for spatial orientation in a blind subterranean mammal. J Exp Biol 208:647–659

Minor comments

Title, and throughout the manuscript: Scientific name should be “Eospalax baileyi

Line 6: This should probably be “Pest Control”.

Lines 19 and 32-33: “and help restore their population”. We cannot know that mole-rats have this motive, so these words should be removed. On line 318 we also find the phrase “ultimately of benefit to the broader population”. Many biologists are wary of what is known as group selection (as opposed to natural selection working to the benefit of individual organisms): I would recommend that the authors reconsider whether they really mean to say this.

Lines 22 and 35: If the idea here is that seismic vibrations are used to detect empty tunnels, this should read “seismic signalling” not “seismic communication” (which requires a receiver animal, as the authors note later).

Lines 25 and 38-39: “when a zokor chose to occupy the empty artificial tunnel”. The signals were recorded prior to occupation and we cannot say when the zokor “chose” anything: I think this would be better rephrased e.g. “before and after occupation of the empty artificial tunnel”.

Lines 40-1: Please add e.g. “(7 out of 8 tunnels)” here so the reader of the abstract knows how many observations contributed to this percentage. “When passing them by” implies that the authors know about the locomotory behaviour of the zokors – is this actually the case? See Major Comment.

Lines 44-6: I think the peak frequency should be higher and the energy lower in the original tunnels, compared to when the artificial tunnels were dug...?

Line 46: “when the zokor detected artificial empty tunnels” should read e.g. “when an artificial tunnel was positioned next to the original”. See Major Comment.

Line 53: “Intraspecific” should be “Conspecific”.

Lines 66-67: “Subterranean rodents are small mammals that live in underground tunnels”. This needs an expanded definition, because many mammals will occupy tunnels periodically for shelter (e.g. foxes, rabbits) but would not normally be referred to as “subterranean”. In fact, a better definition is provided on lines 235-45. This section should be moved from the Discussion and integratd into the Introduction, here.

Lines 73-4: “It is impossible for olfactory or auditory signals to be transmitted through soil between tunnels”. This should be rephrased as it is clearly not “impossible”. The problem is the extreme attenuation, which means that distances over which this is possible would be very limited.

Line 78: “share the information with” should be rephrased to e.g. “the information can be communicated to”

Lines 81-4. The English names here are incorrect and should be changed to the following:

Bathyergus janetta: Namaqua dune mole-rat

Bathyergus suillus: Cape dune mole-rat

Georychus capensis: Cape mole-rat

Tachyoryctes daemon: Demon African mole-rat

Spalax ehrenbergi: blind mole-rat

Line 109: “with 90% of the vegetation coverage” should be “with 90% vegetation coverage”.

Line 112: “Ruthenian medic” appears to be an English name rather than a scientific name. I think the scientific name should be Medicago ruthenica.

Lines 130-1: I do not understand what is meant by seismic signals being “collected every 2m”. Please rephrase and clarify. Please also specify what the distance was between the artificial and natural tunnels, because it is not clear which distance was chosen in the end.

“In nature...in indoor environment” should read “In the natural environment...in the indoor environment”

Figure 2. The words “Plan view” could be put into the caption instead of being repeated. Actually, the cross-section of the tunnel in (b) is not in plan view. Secondly, the cross added to the top and bottom of the original tunnels is slightly confusing because it seems to be separated from those tunnels. Can the figure be adapted to indicate that the cross parts are continuous with the cylindrical parts? Also, please add a scale bar so we can easily see how far apart the tunnels are.

Lines 141, 147: typo in “seismic”.

Lines 148-9: “Among the four geophones, two were placed between the artificial tunnel and original tunnel at a distance of 5 cm from the original tunnel”. The diagrams in figure 2b,c,d show just one geophone between the original and artificial tunnels. Please change the diagram and/or the text to show the true disposition of the geophones.

Also, Figure 2 uses the term “seismometer” while the text uses “geophone”. Please be consistent with this terminology.

Line 150: “reordered” should be “recorded”. Also, “while the plateau zokors detected the empty artificial tunnels” should be removed, since this is an inference and in any case, recordings were made even when there were no artificial tunnels. Instead of this, please specify here how long the recordings lasted for. As I understand it, this is the full 48 hours between the scientists leaving the area and returning to dig up the tunnels.

Given that four geophones were used, how were results from these four combined? What energy level, for example, was chosen, because this will surely differ between these recordings?

Lines 156, 157, 158: three typos in “artificial”.

Line 159: typo in “defended”

Line 162: see Major Comment above about the need for clear definitions.

Lines 164-5: “were used to compare between the original and artificial tunnels” should  I think be changed to “were compared between control and experimental situations”.

Line 167: “firstly” should read “first”.

Line 172: By “with a 95% confidence interval”, do the authors mean e.g. “with a significance threshold set at p=0.05”?

Lines 179-80: It would be clearer to put in numbers of tunnels here, rather than percentages. I am also confused because if seven tunnels were occupied, how can this divide into 67% and 33% (4/7 is 57% and 5/7 is 71%)? Also lines 289-90.

Table 1 should be deleted: the important information is all in the text and the information here is very repetitive. Also, I don’t know what “Layout time” and “number of layouts” mean: what is a “layout”?

Lines 187-8: typo in “activities” and “signals”. Also, I don’t know what is meant by “number of activities” or “distance of seismic signals”. These are not parameters listed on lines 162-165. Should “distance” read “duration”? Should there be a reference here to Fig. 3? The authors say that the differences are signficant: they should provide p-values for each test.

Figure 3. I am concerned about the “duration of seismic signal” for the CK experiments. It looks like the mean duration is very short, and the lower SEM would take that value below zero. Firstly, the value here should be the mean of seismic signals that were actually made, so the authors should be sure that no zero values were included in the average. Secondly, how can the authors be sure that such short-duration signals really were seismic signals generated by the zokors?

Lines 197-8: “A total of 92 series of seismic signals produced by plateau zokors were recorded, including 6 in the original tunnels and 86 in the simulated tunnels”. Are the authors saying that the zokors were physically located in the original tunnels or in the simulate tunnels when these recordings were made? If so, how do they know the location of the animals? Alternatively, might they be comparing control cases (recordings made with no artificial burrow) to experimental cases (recordings made with artificial burrow present)? If so, this needs to be rephrased.

Lines 201-2: “Comparing the parameters of the seismic signals between the original tunnels and when passing by the simulated tunnels”. Same comment as above: should this actually read e.g. “Comparing the parameters of the seismic signals recorded with and without an adjacent artificial burrow...”?

Line 204: “when plateau zokors passed by the artificial tunnel”. How do we know were they were located? Is it just assumed that if a recording was made, the zokors must be passing by?

Line 204-5: “were greater than those in the original tunnel”. Should this read e.g. “were greater than those made in the control experiments without experimental burrows”? Please provide a p-number.

Lines 206, 208, 209, 210, also Fig 4,5 caption: “in the original tunnels”...maybe should be “in the control experiments without experimental burrows” again?  “In the artificial tunnels” should read “in the experiments with artificial burrows”?

Line 207: “Besides” should read “In addition”.

Lines 207-8: “the peak frequency of the main frequency of the seismic signal” should be just “the peak frequency of the seismic signal”?

Figure 4, 5: Should this read “amplitude” rather than “amplification”? Both the x-axis and the y-axis need to be labelled in (b). The units for “energy” need to be included in (c). Also, Fig. 5 refers twice to a “map” but Fig. 4 does not – the terminology used should be the same for both captions so we can see that they may be directly compared. I think “plot” is better than “map”.

Fig. 6: The caption reads “inter-round time interval, number of rounds per series, number of pulses per round, and inter-pulse time interval”. The graphs themselves read “Inter-bout distance”, “pulses per bout”, “bouts per series” and “inter-pulse distance”. Once again, the terminology is different and this makes it really hard to understand what we are looking at. “within the natural and artificial tunnels” should also be rephrased, as I suggest above. “Distances” are apparently measured in seconds here – I presume these should really be “durations” or “intervals”.

Also, it should read “the dot inside the box” since there is only one dot. I don’t understand what “within inner fences” means. I don’t understand what the widths of the violin plots represent. All p-values should be stated.

Discussion: As per the Major Comments, I think that the material in sections 4.1 and 4.2 is really introductory material and should be moved to the introduction. The purpose of the Discussion is to put the results of the current study in context, but all the material in section 4.1 and some of the material in 4.2 is summarising previous studies and answering the questions that I had, as a reader, much earlier in the paper.

Line 248: what does “better” mean, in this context?

Lines 260-1: “The transmission distance of the seismic signals was around 2 m in the field”. Please explain further how this number 2 m is arrived at. Is this the distance at which point the signal is no longer detectable, given ambient seismic noise?

Line 277: “In the present study, it was found that they produced seismic signals while walking inside the original tunnel,” Please explain how it was known that they were producing signals while walking. Can the authors be sure that the animals were not stationary?

Line 280: What is “exploration frequency”?

Lines 282-3: “they increased their staying time at the location in the original tunnel close to the empty tunnel (Fig. 3)”. Figure 3 examines only the number of activites and the duration of the seismic signal. There is no information provided about the location of the zokor: presumably, it could be in the same location but silent, and this would not be detectable in this study. The authors should be very careful to refer to precisely what they were able to measure in statements like this.

Lines 285, 287: Along similar lines to the above comment, the authors describe “the zokor’s detection of the artificial tunnel”... “If the zokor did not receive a response signal from their neighbor, it invaded the empty tunnel and completed the occupation.” These are interpretations made from the changes in seismic patterns, but we cannot know for sure if the zokor detected the tunnel (or maybe just the ground disturbance involved in digging that tunnel), or if the invasion had anything to do with the lack of response signals from a neighbour (since no experiments were made which looked a this). Again, the authors should be precise in distinguishing what were the actual scientific observations, and then what they infer might be happening.

(N.B. The later statement “In these situations, it might be the case that the zokor regarded the tunnel as a potential threat and sought to protect its own territory” is fine. Including “might be the case” makes it clear that this is the authors’ interpretation: this is how the previous passages should have been phrased too).

Line 294: “The receiving of seismic signals in subterranean rodents happens in two ways: via their auditory system and via their somatosensory system”. This is actually controversial. The authors might rephrase this to e.g. “might happen in either or both of two ways”. A recent discussion of this controversy is included as a particular case-study in the book chapter below:

Mason, M.J. & Wenger, L.M.D. (2019) Mechanisms of vibration detection in mammals. In: Biotremology: Studying Vibrational Behavior. Hill, P.S.M. et al. (eds). Cham: Springer.

Lines 294, 306: change “receiving” to “reception”.

Lines 295-6: change “the inner-ear nerve” to “the auditory nerve”.

Line 299: “Some scientists have concluded...”. Please provide references.

Line 308: pleaase change “which is helpful” to “which might potentially be helpful”, because we presumably do not know for sure if this gene product is involved in seismic sensitivity.

Lines 320-1: “plateau zokors can occupy empty tunnels through low-frequency seismic wave signals”. This needs rephrasing to something like “plateau zokors seem to produce low-frequency seismic wave signals as part of the process involved in the occupation of empty tunnels”.

Lines 321-2: “the time taken to explore the surrounding environment increased within the artificial tunnel system”. Do we know how long they took to explore their environment? Or do we just know something about the frequency and duration of seismic signalling. Also, do the authors mean “when an artificial tunnel was present” instead of “within the artificial tunnel system”?

Reviewer 2 Report

This study reports seismic signals produced by a solitary underground rodent before (?) and after the investigators install an unoccupied burrow nearby. The data suggest that the zokors produce seismic signals that allow them to detect the empty tunnels. They then either connect the new tunnels to their own tunnel or else fill the new tunnel with soil to prevent its use. The authors claim that this expanding of home tunnels by occupying vacated tunnels is helpful in restoring populations. This claim does not quite seem logical to this reviewer since an individual animal may gain a little more territory but that does not increase the population. This should be better explained.

This report is somewhat difficult to evaluate because of the lack of clarity, most likely owing to subtleties of English word choice and failure to provide background for some statements. 

For example, the authors in the abstract and elsewhere refer to “restoring” the population, but do not note that there is a need for restoration or how larger territories will accomplish this. 

Specific points:

It would be helpful to indicate the degree of specialization of this species for an underground lifestyle—does it still have vision, for example, or does is spend any part of its life above ground?

Please try to avoid abbreviations throughout except for those that are pretty standard such as units of measurement. Similarly, label the figures with words rather than abbreviations such as figure 2, replace “(a)” with “seismic signals near original tunnels” or something else appropriate. In the figures, it would also be helpful to the reader to spell out other labels such as CT, M, T, S, and the little drawing in the upper right part of figure 2b. Also describe what is meant by a ‘complex’ tunnel. 

Line 148, the placement of two geophones is described, but not the remaining two. Are the geophones different from the seismometers?

Line 161, please define all the terms such as generation time, round, pulses, main frequency, and peak frequency energy.

Table 1 seems to convey little information; the first tunnel was not occupied, but all the later ones were. The percent is not meaningful since there was only one tunnel that was either occupied or not. This could be said in a sentence or two without the table.

Line 188, reference is to signals produced between the zokors, but so far, only single zokors have been mentioned. 

Line 198, were there 6 or 8 original tunnels.

Line 205, If the difference is not significant, it should not be described as greater, but as not reliably different. 

Figures 4, 5. Label the figures with a meaningful title rather than ‘a’ etc. Also Figure 4&5b have no unit label on the x or y axis. 

Figures 6 and 7 are incomprehensible. The whisker plots are too tiny to be easily read but their size could be much greater without increasing the size of the figure itself. The x axis is not described

Line 293, the word “obtain” seems to mean “detect”.This discussion is speculative, but is is not misrepresented, although the point about benefiting the broader population remains unwarranted unless better justified.
